# On the Problem of State Recognition in Injection Molding Based on Accelerometer Data Sets

**DOI:** 10.3390/s22166165

**Published:** 2022-08-17

**Authors:** Julian Brunthaler, Patryk Grabski, Valentin Sturm, Wolfgang Lubowski, Dmitry Efrosinin

**Affiliations:** 1AISEMO GmbH, 4675 Weibern, Austria; 2Linz Center of Meachtronics GmbH, 4040 Linz, Austria; 3Institute of Stochastics, Johannes Kepler University Linz, 4040 Linz, Austria

**Keywords:** injection molding, accelerometer sensor, state recognition, machine-learning algorithm, convolutional neural network

## Abstract

The last few decades have been characterised by a very active application of smart technologies in various fields of industry. This paper deals with industrial activities, such as injection molding, where it is required to monitor continuously the manufacturing process to identify both the effective running time and down-time periods. Supervised machine learning algorithms are developed to recognize automatically the periods of the injection molding machines. The former algorithm uses directly the features of the descriptive statistics, while the latter one utilizes a convolutional neural network. The automatic state recognition system is equipped with an 3D-accelerometer sensor whose datasets are used to train and verify the proposed algorithms. The novelty of our contribution is that accelerometer data-based machine learning models are used to distinguish producing and non-producing periods by means of recognition of key steps in an injection molding cycle. The first testing results show the approximate overall balanced accuracy of 72–92% that illustrates the large potential of the monitoring system with the accelerometer. According to the ANOVA test, there are no sufficient statistical differences between the comparative algorithms, but the results of the neural network exhibit higher variances of the defined accuracy metrics.

## 1. Introduction

In the last few decades, there has been an increasing interest in the application of big data and smart technologies in industry to guarantee the specified production quality and efficiency, as well as to track various anomalies during a manufacturing process. Injection molding production also requires state-of-the-art technology for process monitoring, data collection and their classification. Injection molding is known to be a low-cost, high-capacity production technology in the plastics industry. It is usually a process which involves injecting molten polymer materials from a heated barrel cylinder into a closed mold, followed by cooling of the plastic product and then pushing it out from the mold. Although the process is fully automated, there is a need for some auxiliary monitoring system to monitor the production process with the aim to increase the efficiency and to reduce costs.

Despite some advances in injection molding production quality control, no universal automatic methods have yet been proposed to recognize real running-time and down-time periods. The ability to track this information would obviously improve the economic and environmental efficiency of this type of production. We develop the monitoring system equipped with a 3D-accelerometer sensor on the injection molding machine for this purpose. The time-ordered datasets are recorded by the sensor in the form of three-dimensional vectors. Then, they are used for time series classification to recognize such individual states as mold closing, injection unit actions, plasticizing and demolding. In contrast to the previous results, our analysis includes the following contributions. We develop two state recognition algorithms based on supervised paradigms. In the first case, the relatively small amount of labeled data are used to estimate threshold levels for the specified features of descriptive statistics. In the second case, a convolutional neural network is implemented directly to the datasets recorded by a monitoring system. The real data are used for algorithm training and verification. Different accuracy metrics of the classification models are evaluated. To determine whether one learning algorithm outperforms another, the ANOVA-test is used. In a post-processing procedure, it is determined whether the sequence of recognized cycle steps represents a correct injection molding cycle. This information is used to determine whether or not production is to take place. This post-processing procedure is not the subject of this publication. This publication deals only with the recognition of cycle steps in acceleration data.

The remaining chapters of this paper are structured as follows: first, the related work is presented in Section 2. Section 3 describes which data are analyzed and how they are obtained. After the specification of the pre-processing, the two classification algorithms presented in this paper—a tree based approach and an artificial neural network model—are described in Section 4. More information about the metrics that are used in the numerical analysis of these two models as well as the evaluation itself can be found in Section 5. Finally, the outcome is summarized and discussed in Section 6.

## 2. Related Work

Multiple publications discuss the topic of sensor technology and machine learning to support injection mold manufacturing. A main topic concerned in literature is the quality prediction and assessment using machine learning techniques; see, for instance, [1,2,3,4]. Another useful approach is the utilization of a simulation model or digital twin and machine learning to support manufacturing processes as presented in [5,6,7].

In recent years, multiple schemes employing sensor data for production support and quality control were proposed in literature, such as pressure [8,9,10,11,12] and temperature [8,9,10,11] or ultra-sonic measurements [13]. A review on different sensor types utilized in-mold for process control can be found in [14].

In [15], the authors present a comprehensive review on the application of multiple supervised and unsupervised machine learning techniques for monitoring injection mold processes. An assistance system for quality assurance in the plastic injection molding process was described in [16]. They used pressure, force and temperature sensors which were placed inside the tool cavity. A support vector machine was then used for the classification into good and bad parts. The authors of [17] presented an approach to include machine learning in an injection molding process with the goal of enhancing the production performance. In their study, they used a wide range of sensors attached to the tool. With these measurements, a machine learning model was trained to predict the best process parameters for the injection molding machine. Several machine learning algorithms were utilized in [18] to test and compare performances in quality prediction by means of main features of time and temperature by injection molding. The problem of the quality prediction was analysed in [19], where convolutional neural networks were trained on thermographic images. Among others, we can mention also the works [20,21], where the quality classification and anomaly detection in injection molding were discussed in the framework of different aspects.

Accelerometers have long been used to track the state of moving objects; see, for example, [22] and the literature overview with references therein, where the low frequency acceleration data were used for feature selection and classification in a human activity recognition problem. The authors used 50 Hz sensor data from a triaxial acceleration sensor; in addition, they also used data from a gyroscope sensor. From these data, they calculated 585 features, which where then scored by three different feature selection methods. To get the optimal feature set a classifier was trained several times with different number of features. The results show that about 40 features are enough to get the best classification performance.

To the best of our knowledge, there are very few papers dedicated to utilization of accelerometers to monitor the injection molding process. The authors of [23] proposed a monitoring scheme for an injection molding tool, which consists of a pressure and acceleration sensor directly integrated into the injection tool. The authors conclude that “a reduction in the maintenance costs of the injection tool is expected”. The monitoring system proposed in paper [24] uses an accelerometer to record injection molding vibration signals. These values were then used as input variables for logistic modelling to predict the mold flash, which is an excess plastic that forms on the surface of injection molded parts.

## 3. Data

This paper is based on the recording of acceleration data from injection molding machines as they occur in a plastic manufacturing facility. The left part of Figure 1 shows such an injection molding machine, which was also used for the purpose of data collection for this work. The data are measured by an triaxial accelerometer with a sampling frequency of 100 Hz, which means that a sample is recorded every 10 ms. The sensor is permanently mounted on the injection molding machine and records the acceleration for each of the three mutually orthogonal *x*, *y* and *z*-axes. To guarantee reliable data recording, it is attached in a tamper-proof manner so that slippage is not possible, as shown on the right in Figure 1 where the sensor is mounted on the machine. On the right side of Figure 1, one can see the mounted acceleromoter. The left picture shows the whole injection molding machine, and the right circle shows the position of the sensor.

Appropriate and exact labels for the acceleration data are necessary for training and evaluation. These are read out synchronously from the programmable logic controller of the injection molding machine. In total, there are five different states representing the current process step of the machine that can occur: closing, injection unit, plasticizing, demolding and null state—where the null state describes the state in which none of the other four states is active.

The experiments presented in this paper used production data from the manufacture of 15 different products recorded on five different injection molding machines. All machines are from the Victory series of the Austrian machine manufacturer Engel. The years of construction of the five machines used range from the mid-1990s to the present.

We can see in Figure 2 that the distribution and structure of states varies greatly between different data sets. In particular, the difference in the realizations can be seen most clearly in the plasticizing state. In (a) and (b), the energy in the signal is significantly lower than in (c).

One interesting point is to use acceleration data to identify when the machine is producing and when it is not. For this purpose, we introduce the terms producing and non-producing. Producing means that the states closing, injection unit, plasticizing, demolding and closing occur one after the other. This sequence of states suggests that a plastic part has been produced, and an injection molding cycle has been completed—a different sequence of states is evaluated as non-producing. In addition, if the null state is active for too long between two cycles, this is considered non-producing.

## 4. Data Analysis

### 4.1. Pre-Processing

Besides the vibration of the machine, the sensor also detects a constant acceleration due to gravity. To eliminate this effect, the first step is to subtract a running mean from each axis. Afterwards, the magnitude of the acceleration is calculated:(1)$$\begin{array}{c} M\left(i,n\right)=\sqrt{{\left(x_{i}-\sum_{j=-n}^{n}\frac{x_{i+j}}{2n+1}\right)}^{2}+{\left(y_{i}-\sum_{j=-n}^{n}\frac{y_{i+j}}{2n+1}\right)}^{2}+{\left(z_{i}-\sum_{j=-n}^{n}\frac{z_{i+j}}{2n+1}\right)}^{2}}, \end{array}$$
for i=n+1,n+2,…, where $\left(x_{i},y_{i},z_{i}\right)$ denotes the respective values of time series at time *i*. An example of the original and resulting time series for n=100 is depicted in Figure 3 below:

The resulting time series with elements (Equation 1) has more contrasting segments characterised by low and high variances which is clearly an advantage for further classification tasks.

### 4.2. Tree Based Approach

When selecting possible methods for analyzing the data, it was necessary to use automatable machine learning methods. The reason for this is that it is assumed that the underlying data lake will become larger and larger. It is assumed that, in the future, it will be necessary to repeatedly train new models over an ever larger data set. For such adaptations to be possible in an efficient manner, it is necessary to rely on machine learning. Within the machine learning methods, two established methods, namely a tree-based approach and an artificial neural network, were chosen because they are established and well applicable methods.

For our first approach, we extract 44 statistical features from the pre-processed magnitude of acceleration. Twenty-two of these features stem from the original time-series, and 22 from the first order differences of our signals, which may be interpreted as an estimator for the jerk or jolt, which is defined as the first derivative of the acceleration. To this end, we take a running window of length 200 ms for which we calculate said features. The features are then labeled with the class label positioned at the 100th time-step in such an interval. The resulting feature set is then under-sampled, which is achieved by randomly sampling from each class to attain equal class frequencies. Finally, we apply a gradient boosted tree algorithm on this reduced set which we then use to predict the labels of the yet unseen current test set.

### 4.3. Artificial Neural Network

In our neural network approach, we were influenced by the idea of semantic segmentation, a well-known computer vision problem in which each pixel of an image is assigned a label. In the context of time series data, this means that we predict a label for every timestamp in the input series. To achieve this, we implement a 1D fully-convolutional network, this allows us to get a prediction for each input value. As the time duration of the states can vary greatly, we decide to counteract this problem by applying convolutions with various receptive fields to the same input time series. This idea is based on the work presented in [25]. The key difference is that we adapt the architecture in such a way that it predicts a label for every timestamp in the input series. This is a significant change that allows 1D convolutions to be used for segmenting time series data. In Figure 4, a sketch of one such convolutional block is shown.

The neural network model can be interpreted as a function, $g:R^{MxT}\to R^{KxT}$, where *T* indicates the length of the input time series, *K* the number of different classes and *M* represents the number of input values per time stamp. In our case, M=4 consists of the acceleration data of the three axes as well as their magnitude. Since we are dealing with a multi-class problem, the cross entropy loss function is used as objective function for the training. For the setting in this paper, the cross entropy error for one prediction is depicted in equation (Equation 2), where ${\hat{Y}}_{n}\in R^{K\times T}$ is the model prediction and $Y\in R^{K\times T}$ the corresponding label:(2)$$CE\left(Y,\hat{Y}\right)=-\sum_{t=1}^{T}\sum_{c=1}^{K}y_{c,t}\cdot log\left({\hat{y}}_{c,t}\right)\;,\;\mathrm{with}\;{\hat{y}}_{c,t}\in \left[0,1\right]\;\mathrm{and}\;y_{c,t}\in \left\{0,1\right\}.$$

In order to minimize the cross entropy loss, we use the *Adam* [26] optimizer with a learning rate of 2e−4.

## 5. Results and Numerical Analysis

This section describes the two experiments carried out in this work. In the first sub-section, the used metrics are described. After that the two different experimental settings, including their results, are presented. For the considered states which are described in Section 3, we use the following abbreviations:1→Closing,2→InjectionUnit,3→Plasticizing,4→Demolding,5→NullState

### 5.1. Metrics

The choice of the right performance measurement is a crucial task in a machine learning project. For classification problems, the accuracy is an often used metric because of its simplicity and interpretability. Besides the accuracy, there are plenty of other metrics which can be used to evaluate and analyse classification results. In [27], the most common performance measures are described and compared. To assess the classification quality, we resort to confusion matrices and different performance metrics. A confusion matrix [28] *C* of an algorithm *A* on some dataset D consists of entries $C_{i,j}$, which are defined as “Number of examples of class *i* in D classified as *j* by *A*”. As our matrices contain 25 entries each, we also state multiple performance metrics, which can be interpreted as a function of the entries of a confusion matrix and concisely represent the classification quality. We adjust the confusion matrices presented is this article such that they are rescaled for every true state, i.e., every row adds up to 100 (percent). Since we are confronted with an imbalanced class-distribution, we decided that, for balanced accuracy (BACC), as defined in (Equation 3) below, as an internal performance metric during the training of classical algorithm:(3)$$\begin{array}{cc} BACC(C) & =\frac{1}{5}\sum_{i=1}^{5}\frac{C_{i,i}}{\sum_{j=1}^{5}C_{i,j}} \end{array}$$(4)$$\begin{array}{cc} ACC(C) & =\frac{\sum_{i=1}^{5}C_{i,i}}{\sum_{i=1}^{5}\sum_{j=1}^{5}C_{i,j}} \end{array}$$(5)$$\begin{array}{cc} F-Score(C) & =\frac{1}{5}\sum_{i=1}^{5}\frac{2C_{i,i}}{\sum_{j=1}^{5}C_{i,j}+\sum_{j=1}^{5}C_{j,i}} \end{array}$$

Since we employed a cross-validation scheme, we add all confusion matrices for a final resulting matrix, for which we calculated and present the results in the following section.

### 5.2. Numerical Analysis

In the present work, we employ two evaluation methods for our proposed systems. The first one, called Inter-tool validation, always splits the data in such a way that all data from one tool is either in the train dataset or in the test data set. For the second experiment, called Intra-tool validation, the database is randomly split into a train dataset and into a test data set. With these evaluation methods, we are able to analyse if our methods learn injection moulding process specific characteristics or if only tool dependent ones are learned.

#### 5.2.1. Experiment I: Inter-Tool Validation Results

The available database consists of data from 15 different injection tools. In this experiment, a five-fold cross-validation is performed in the manner that all data from exactly three tools are used as the test data and the remaining data for training. Below, the summed up confusion matrices, as well as the averaged metrics, are presented.

The inspection of our two confusion matrices depicted in Table 1 and Table 2 implies that both algorithms tend to confuse state 2 (Injection Unit) and state 5 (Null State). This may be attributed to the general nature of our Null State and the rather brief occurrences of injection.

In addition to these evaluations, we also conducted a statistical evaluation to assess whether there is a statistically significant difference between our two chosen approaches. To this end, we depicted three performance measures as box-whisker charts and compared the values of all 15 datasets using a classic one-way ANOVA with repeated measures.

From Figure 5, Figure 6 and Figure 7, the similar performance of our two methods when considering our three performance measures is apparent. In Table 3, we present results from conducting a two-sided test for equal means, which corroborates with the impression of similar performance. What we can infer from Figure 5, Figure 6 and Figure 7 is that the highest and lowest values of each performance measure are attained by the neural network approach.

The high *p*-values presented in Table 4 do not indicate any significant difference in all performance measures.

#### 5.2.2. Experiment II: Intra-Tool Validation Results

Our second experiment considers a different data split up, such that we have each machine–tool combination already in the data set. To this end, we split up the data into training and test data, where the test data consist of 20% of each of our 15 data sets. This split up is repeated five times, and both algorithms are learned on each training set and applied on each test set, respectively, as described in in the introduction to Section 5.2.

From inspecting the two confusion matrices in Table 5 and Table 6 and the performance measures in Table 7, it is evident that including data from a machine on which the algorithms are tested into the training set significantly increases the classification quality. When comparing the results from our tree classifier with the performance in Experiment I, we can see that the classification error (1-ACC) reduces from 0.3154 to 0.1973, a reduction of about 37.45%. The same comparison for our NN approach yields an error reduction from 0.3302 to 0.0834, a decrease of about 74.74%. Moreover we can observe that, while, for the results from Experiment I, the performance values are rather similar, in our Experiment II, the respective values differ greatly. The Neural Network attains an average BACC value of 0.9413, while the tree algorithm achieves 0.8571. In terms of error, this translates to an error that is 58.92% lower for our neural network. In case of ACC, we also observe an error reduction of more than 50%. These observations suggest that the generalisation power of our algorithms to unseen machine/tool combinations still has optimization potential. We assume that a larger and even more diverse data-base would act as a partial remedy.

## 6. Conclusions

This paper has proposed a monitoring system for classifying five different states in the injection molding process. We evaluated two different machine learning approaches, for which we recorded acceleration data using a sensor attached to the injection molding machine. The corresponding labels for the model training were obtained from the programmable logic controller. Both methods were evaluated in two different experiment settings. In the first experiment, the training and test data differ in the way that the test dataset contains injection moulding cycles of tools that were not included in the training data. For this setting, both methods perform quite similarly with a balanced accuracy of 72% and 74%. The main difference is that the results from the neural network approach show higher variance. This is due to the fact that the acceleration data have very different characteristics regarding the used tools the sensor was mounted on. In the second experiment, the collected datasets from all tools were combined and then randomly split into a train and test set. Here, the performance of both models increases up to 86% and 94%.

These results show that there is a high potential in detecting various states in the injection moulding cycle. Nevertheless, the results also suggest that the proposed methods still show a lack of generalization. In order to improve this weakness we suggest using a larger dataset with a higher diversity of injection tools.

## Figures and Tables

**Figure 1 sensors-22-06165-f001:**
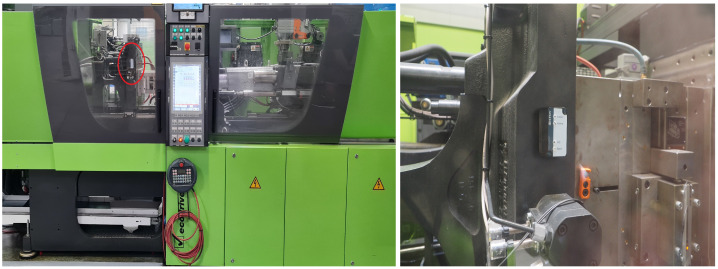
The left image shows an injection molding machine with a mounted 3-axis accelerometer (sensor marked by a red circle in the center of the image). The right picture shows the mounting of the same sensor in detail.

**Figure 2 sensors-22-06165-f002:**
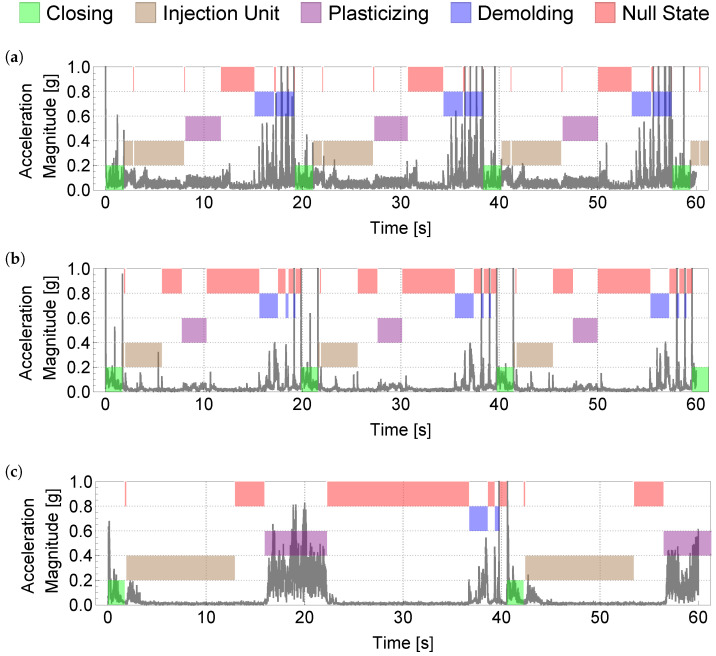
Depiction of excerpts from three different datasets (**a**–**c**). The injection molding cycles shown in (**a**,**b**) are rather short cycles of slightly more than 20 s, (**c**) shows a slightly longer cycle of about 40 s in length.

**Figure 3 sensors-22-06165-f003:**
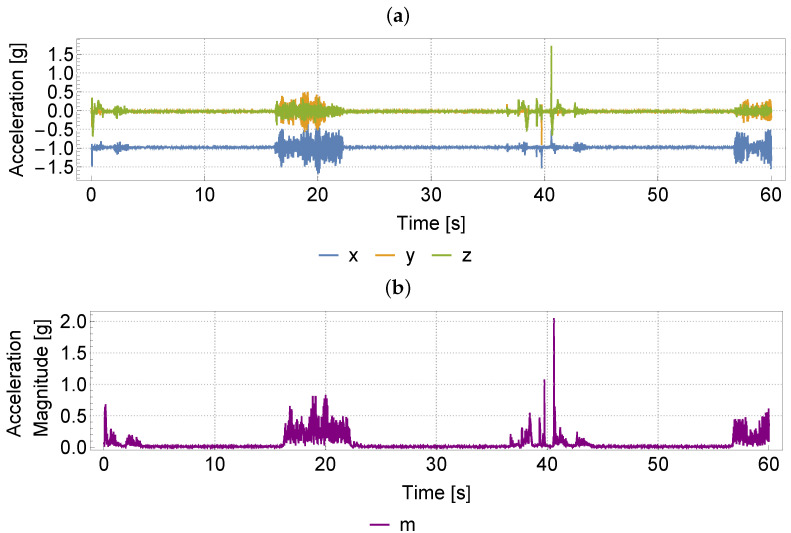
Depiction of the pre-processing step: (**a**) original 3D data set; (**b**) transformed 1D time series acceleration magnitude M(·,100).

**Figure 4 sensors-22-06165-f004:**
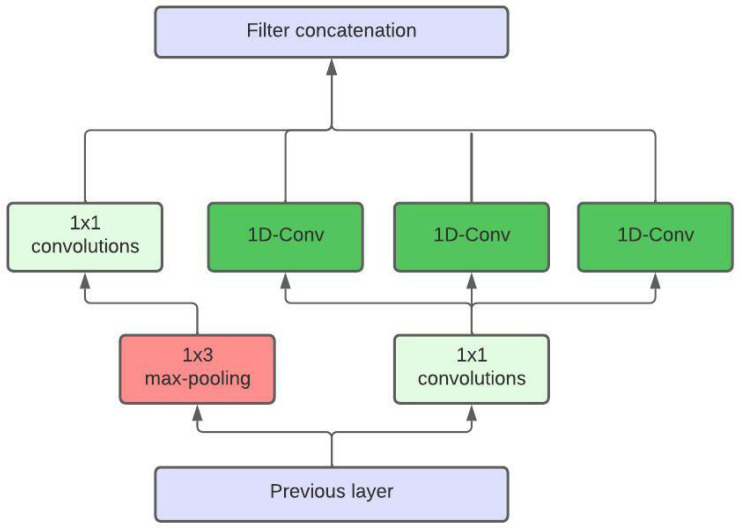
Our Inception module for the time series segmentation. Note that the three 1D-Conv blocks marked in green have different receptive fields.

**Figure 5 sensors-22-06165-f005:**
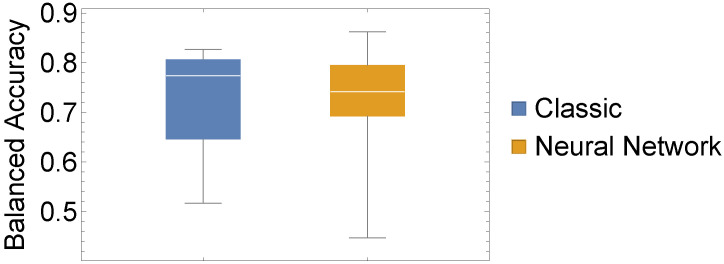
Experiment I: Box whisker chart of BACC values of our 15 datasets for both considered methods.

**Figure 6 sensors-22-06165-f006:**
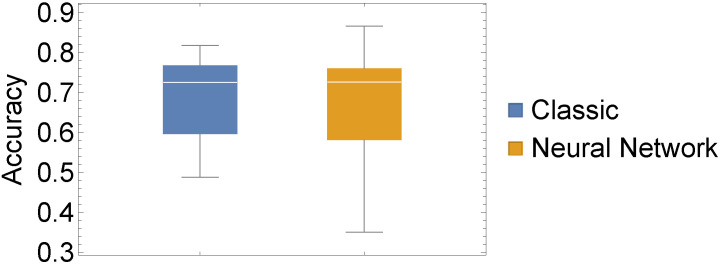
Experiment I: Box whisker chart of ACC values of our 15 datasets for both considered methods.

**Figure 7 sensors-22-06165-f007:**
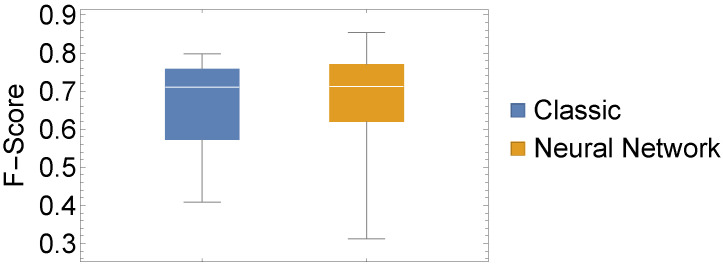
Experiment I: Box whisker chart of F-Score values of our 15 datasets for both considered methods.

**Table 1 sensors-22-06165-t001:** Experiment I: Confusion matrix of our tree-based algorithm.

		Predicted Labels				
		1	2	3	4	5
True Labels	1	84.6	2.60	0.106	12.4	0.229
	2	8.13	53.9	8.49	0.518	28.9
	3	0.217	14.4	79.6	4.37	1.40
	4	15.8	0.368	0.0124	81.0	2.77
	5	2.64	18.9	3.56	10.6	64.2

**Table 2 sensors-22-06165-t002:** Experiment I: Confusion matrix of our neural network.

		Predicted Labels				
		1	2	3	4	5
True Labels	1	84.8	1.95	3.82	7.71	1.68
	2	1.27	64.4	8.92	1.20	24.2
	3	10.7	8.60	78.5	0.965	1.18
	4	7.55	1.21	0.353	88.4	2.52
	5	2.30	35.2	4.42	3.65	54.4

**Table 3 sensors-22-06165-t003:** Summary of results from our Experiment I.

Model	BACC	ACC	F-Score
Classic	0.7268	0.6846	0.6829
NN	0.7411	0.6698	0.7002

**Table 4 sensors-22-06165-t004:** Statistical comparison of our results based on performance measures for each data-set.

Perf Measure	Balanced Acc.	Accuracy	F-Score
* **p** * **-value**	0.9871	0.7837	0.9281

**Table 5 sensors-22-06165-t005:** Experiment II: Confusion matrix of our tree-based algorithm.

		Predicted Labels				
		1	2	3	4	5
True Labels	1	94.7	1.34	0.651	3.83	0.0385
	2	3.39	72.1	1.50	0.0316	23.0
	3	0.0524	2.96	96.0	0.123	0.854
	4	3.50	0.221	0.	94.6	1.65
	5	2.18	12.5	4.21	10.0	71.1

**Table 6 sensors-22-06165-t006:** Experiment II: Confusion matrix of our neural network.

		Predicted Labels				
		1	2	3	4	5
True Labels	1	98.8	1845	524	1053	1415
	2	0.231	85.7	0.412	0.121	13.5
	3	0.0705	0.337	99.1	0.0303	0.492
	4	0.187	0.133	0.0286	98.9	0.714
	5	0.321	10.1	0.755	0.761	88.1

**Table 7 sensors-22-06165-t007:** Summary of results from our Experiment II.

Model	BACC	ACC	F-Score
Classic	0.8571	0.8027	0.8152
NN	0.9413	0.9166	0.9338

## Data Availability

The authors can be contacted to obtain data used in the study.

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
