# Peer review of "On the Problem of State Recognition in Injection Molding Based on Accelerometer Data Sets"

_sensors, 2022, doi:10.3390/s22166165_

Round 1

Reviewer 1 Report

The paper proposes a numerical solution for the state recognition in injection molding based on accelerometer data sets.

Although the results and numerical Analysis part are well presented, the paper needs to be improved with a substantial part regarding related works. The lack of this analysis regarding the literature is also reflected in the small number of references.

Eventually divide the introduction section into 2 and the related work part to be revised.

Author Response

Dear Reviewer,

Thank you for pointing this out.

We have followed your advice and divided the introductory chapter into two chapters (Introduction and Related Work).

In addition, we have extended the section on related work and thus also significantly increased the number of references.

kind regards

Julian Brunthlaer

Reviewer 2 Report

The Paper has been written in a good way. However,  I recommend the following comments to improve the Paper: 

1. The novelty and originality of the Paper are not apparent. 

2. The authors did not discuss how they get data from sensors. 

3. There is not any foto of the experimental set-up. Also, detail of the experiment should be added to the Paper. 

4. It would be better if results were presented in a normalized format, which makes it easy to compare. 

5. The authors choose the simple method of neural network. Would you please provide the reason and also is it possible to use other methods. 

Author Response

Dear Reviewer,

we would like to thank you very much for studying our draft and for your constructive comments.

I would like to address the points individually:
1) Novelty: We have added a novelty statement in the abstract.
2) Data acquisition: Additional explanations have been added in the third chapter, which deals with data collection and shape.
3) Photo: Two photos of the experimental setup were added.
4) Normalization of the confusion matrices: Following the suggestion, the confusion matrices were normalized according to the true values.
5) Reasoning for the choice of methods: In the fourth chapter, which deals with the two methods whose results are presented, an introductory paragraph was added explaining the necessity of using machine learning methods and the specific selection of the two methods.

kind regards

Julian Brunthaler

Round 2

Reviewer 1 Report

The authors took into account the observations made and substantially improved the work.